# Dual function of HPF1 in the modulation of PARP1 and PARP2 activities

Tatyana A. Kurgina [1,2,3], Nina A. Moor[1,3], Mikhail M. Kutuzov[1], Konstantin N. Naumenko[1], Alexander A. Ukraintsev[1] & Olga I. Lavrik [1,2✉]

Poly(ADP-ribosyl)ation catalyzed by poly(ADP-ribose) polymerases (PARPs) is one of the immediate cellular responses to DNA damage. The histone PARylation factor 1 (HPF1) discovered recently to form a joint active site with PARP1 and PARP2 was shown to limit the PARylation activity of PARPs and stimulate their $NAD^+$-hydrolase activity. Here we demonstrate that HPF1 can stimulate the DNA-dependent and DNA-independent autoPARylation of PARP1 and PARP2 as well as the heteroPARylation of histones in the complex with nucleosome. The stimulatory action is detected in a defined range of HPF1 and $NAD^+$ concentrations at which no HPF1-dependent enhancement in the hydrolytic $NAD^+$ consumption occurs. PARP2, comparing with PARP1, is more efficiently stimulated by HPF1 in the autoPARylation reaction and is more active in the heteroPARylation of histones than in the automodification, suggesting a specific role of PARP2 in the ADP-ribosylation-dependent modulation of chromatin structure. Possible role of the dual function of HPF1 in the maintaining PARP activity is discussed.

[1] Institute of Chemical Biology and Fundamental Medicine, SB RAS, Novosibirsk, Russia. [2] Novosibirsk State University, Novosibirsk, Russia. [3] These authors contributed equally: Tatyana A. Kurgina, Nina A. Moor. ✉email: lavrik@niboch.nsc.ru

Poly(ADP-ribosyl)ation (PARylation) is a dynamic post-translational modification of biomolecules that plays an indispensable role in regulating a number of biological processes, including DNA damage response. Poly(ADP-ribose) (PAR) composed of linear and/or branched repeats of ADP-ribose is synthesized by enzymes called poly(ADP-ribose) polymerases (PARPs) that utilize $NAD^+$ as a substrate to modify themselves (automodification) (Fig. 1a, b) or target molecules (heteromodification)[1,2] (Fig. 1c, d).

The best studied PARP family member, PARP1, is the abundant nuclear protein involved in multiple cellular processes, among them, DNA repair regulation. PARP1 serves as a sensor of the DNA lesions usually induced by ionizing irradiation and oxidative stress and signals to recruit the appropriate proteins to the sites of damage[3,4]. PARP2 was also discovered as an enzyme that catalyzes the synthesis of PAR[5]. The role of PARP2 and its cooperation with PARP1 is under intensive investigation[6–10]. In human cells, the majority of PARP activity is exerted by PARP1 (around 90%) and by PARP2 (10-15%)[7]. It is known that neither PARP1 nor PARP2 is required for viability in mice, but parp1$^{−/−}$parp2$^{−/−}$ double knockouts are embryonic lethal with considerable genomic instability[11]. PARP2 shares significant homology with PARP1 in the catalytic domain structure, but differs in the domain architecture due to the absence of zinc fingers and BRCT domain[7]. Roles of PARP1 and PARP2 in base excision repair and single-strand break repair were intensively studied[12–14]. Overlapping functions of PARP1 and PARP2 in regulation of these processes were shown by using model DNA duplexes[15,16] as well as nucleosomes[17,18]. PARP2 is comparable with or even more efficient than PARP1 in binding DNA breaks, but has a lower affinity for intact DNA and AP sites[8,16,19]. Difference in the affinities of PARP1 and PARP2 for various DNA structures in the nucleosome context was lately revealed[18]. PARP2 synthesizes shorter PAR chains than PARP1 does and also inhibits PARP1-catalyzed PAR synthesis via hetero-oligomerization of the two PARPs[8,10]. It has been shown that removal of histone H3 is largely suppressed in PARP2-deficient cells[20]. Thus, PARP1 and PARP2 can perform both overlapping and specific functions in maintaining genome stability, and these enzymes cooperate in the regulation of cellular processes.

Recently the new histone PARylation factor (HPF1) modulating activity of PARP1 and PARP2 was discovered[21,22]. HPF1 was shown to complete the active site of these enzymes via complex formation, and the role of the joint active site was suggested to switch the PARylation specificity to serine residues[23]. HPF1 plays an essential role in the PARP1- and PARP2-catalyzed PARylation of histones[22,24,25]. It was proposed that HPF1 binds only the DNA-activated form of PARP1/PARP2 because the auto-inhibitory helical domain (HD) when bound to the ADP-ribosyl transferase (ART) subdomain of the catalytic (CAT) domain prevents complex formation of HPF1 with PARP1/PARP2[22,26–29]. The conformational change of HD induced by DNA/nucleosome binding promotes the interaction of PARP1/PARP2 with HPF1[27,30]. It is interesting that shortening of the synthesized PAR chains was detected in the presence of HPF1[22]. The authors explained this effect by shielding of amino acid residues important for PAR chain elongation (His826 and His381 in PARP1 and PARP2, respectively) in the complex with HPF1. In addition to affecting the substrate specificity and the length of polymer chain in the ADP-ribosylation reaction, HPF1 was shown to enhance $NAD^+$-hydrolysis catalyzed by PARP1[31]. Recently it was shown that initiation and elongation steps of ADP-ribosylation are distinctly regulated by HPF1, ARH3, and PARG hydrolases, and that the HPF1-dependent initial attachment of ADP-ribose to Ser residue can be elongated by PARP1 alone[32].

PARP1 and PARP2 are known as therapeutic targets for the treatment of malignant tumors and other diseases. Several inhibitors of PARP are already using as anticancer drugs. Cells with knockout of HPF1 revealed high sensitivity to PARP inhibitors[21]. Rudolph and coauthors from Luger's group showed that HPF1 differently modulates affinity of some PARP inhibitors for the PARP1-nucleosome complex, but this effect does not extend to PARP2[33]. Therefore, the comparative study of the mechanism of HPF1-dependent modulation of the activities of PARP1 and PARP2 is of particular interest.

In the present study, we compared effects produced by HPF1 on the PARP1 and PARP2 activities in the autoPARylation reaction performed in the presence of DNA or nucleosome, as well as in the heteroPARylation of histones. HPF1 was found to stimulate the automodification of both PARP1 and PARP2 in a concentration-dependent manner, with the effects detected for PARP2 exceeding those for PARP1. Under optimal HPF1 concentration (produced the highest stimulatory action) no enhancement of $NAD^+$-hydrolase activity of PARPs was detected. PARP2 was revealed to catalyze more efficiently modification of histones than the automodification.

## Results

**HPF1 enhances PARP1 and PARP2 automodification concomitantly with the switch of ADP-ribosylation onto histones.** Influence of HPF1 on the activities of PARP1 and PARP2 in the automodification reaction and heteromodification of histones was tested at varied concentrations of HPF1. To activate PARP1 and PARP2, a model 147-mer double-stranded DNA and the nucleosome core particle (NCP) constructed from this DNA were used. The products of PARylation reaction were analyzed by SDS-PAGE separation and quantification, using [$^{32}$P]$NAD^+$ (Fig. 2a,

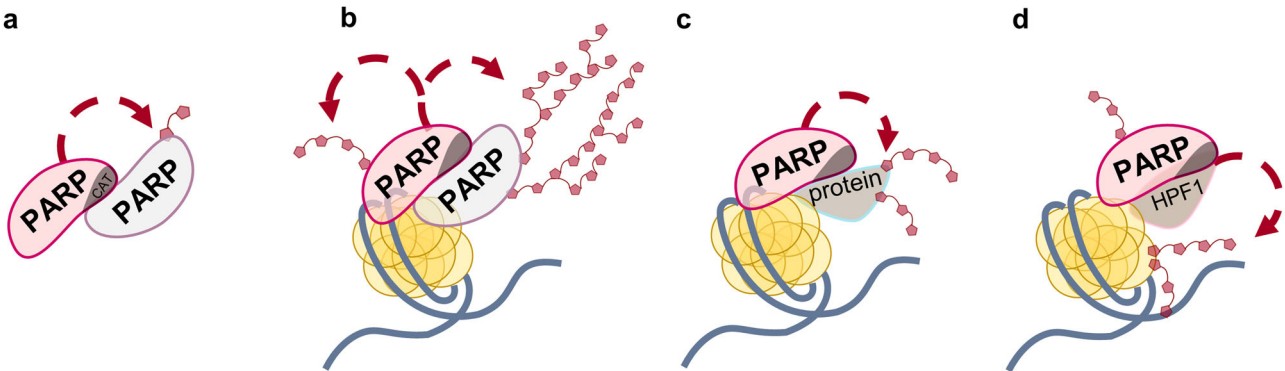

**Fig. 1 Different catalytic activities of PARPs. a** The basal (DNA-independent) activity; **b** the DNA-dependent automodification suggested to occur in *cis* and *trans*[40]; **c** the DNA-dependent heteromodification of target protein[40]; **d** the HPF1-induced heteromodification of histones.

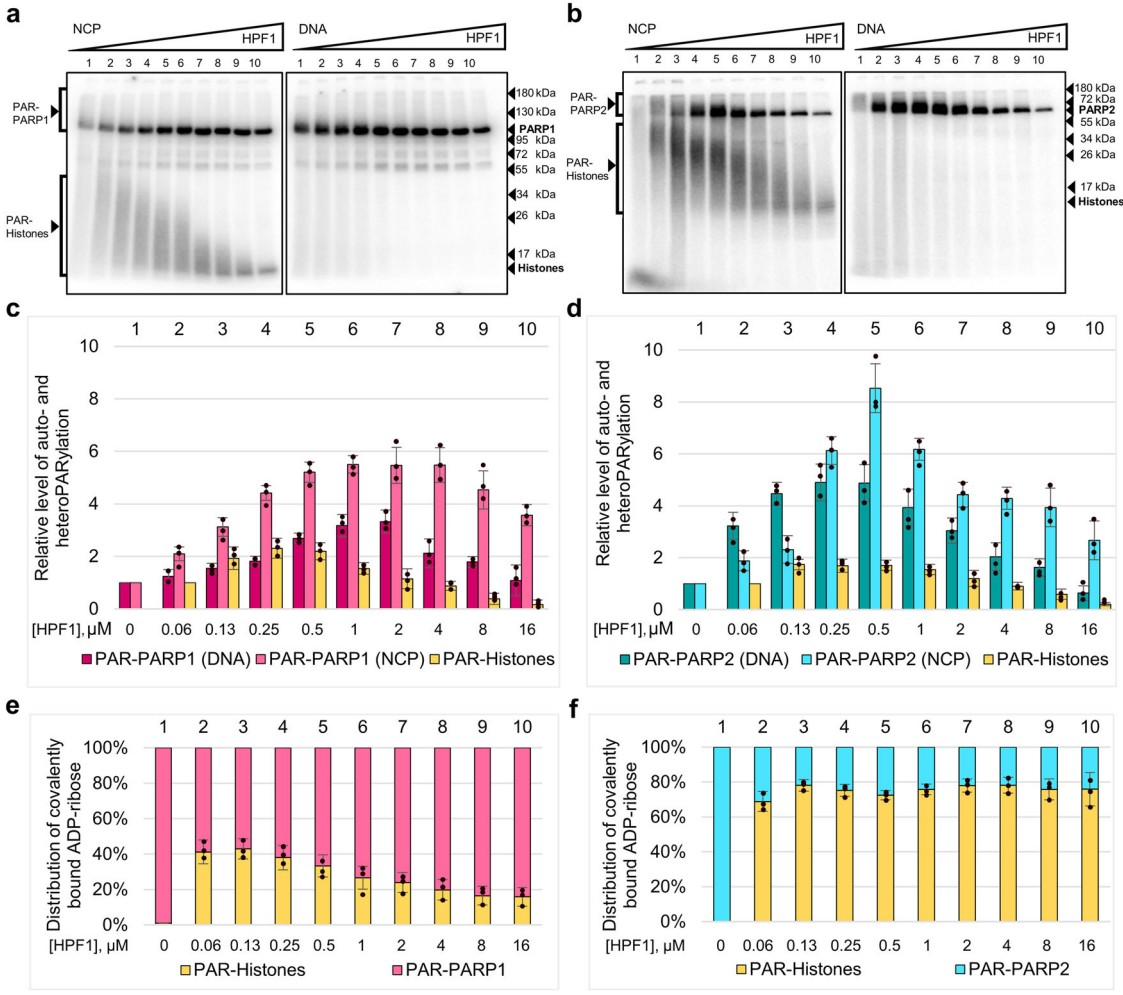

**Fig. 2 HPF1 stimulates the autoPARylation of PARP1 and PARP2 and heteroPARylation of histones.** Covalent binding of $^{32}$P-labelled ADP-ribose to proteins was performed by incubation of PARP1 (**a**) and PARP2 (**b**) (500 nM) with [$^{32}$P]NAD$^+$ (1 µM), in the presence of DNA (250 nM) or NCP (250 nM) (as specified above the autoradiograms) and increasing concentrations of HPF1 (from 60 nM to 16 µM); the products were separated in 10% (**a**) and 20% (**b**) SDS-PAG. The positions of PARylated proteins and their native forms (and molecular weight markers) are indicated on the left and right sides of the autoradiograms. **c**, **d** Histograms present relative labelling levels (the mean ± SD of three independent measurements and individual data points) of PARP1, PARP2, and histones determined by normalization of the ADP-ribose amounts covalently bound to the given protein in the presence of HPF1 (samples 2–10 for PARP1/PARP2 and samples 3–10 for histones) to the respective amount in the absence of HPF1 or its minimal concentration (sample 1 for PARP1/PARP2 and sample 2 for histones). **e**, **f** Histograms show distribution (%) of the total amount of covalently bound [$^{32}$P]ADP-ribose in each sample between the modified PARP1(PARP2) and histones.

b). The presence of HPF1 even at a limited concentration (60 nM in sample 2 compared to 500 nM of PARP1/2) was detected to stimulate PARylation of histones. The increasing concentrations of HPF1 affected the total ADP-ribosylation level of PARPs and histones as well as the length of ADP-ribose polymer attached to the proteins. The heterogeneous mixtures of modified proteins with quite different electrophoretic mobilities (visualized as smearing) became more uniform in the presence of HPF1 as shown by the tight intense band with the increased electrophoretic mobility corresponding to the oligo/mono(ADP-ribosyl)ated proteins. The stimulatory effects produced by HPF1 on the automodification of PARP1/PARP2 were more pronounced when nucleosome instead of DNA was used for the PARP activation. The maximal effects produced by HPF1 on the PARP1-catalyzed automodification in the presence of nucleosome (a 6-fold increase) and DNA (a 3-fold increase) were observed at a 2-fold excess of HPF1 over PARP1 (Fig. 2c). The level of PARP2 automodification was enhanced to the maximal extent (8- and 5-fold in the presence of NCP and DNA, respectively) by the

equimolar HPF1 concentration (Fig. 2d). When HPF1 concentration was further increased, its stimulatory effect on the automodification of both PARP1 and PARP2 decreased.

The concentration dependences of effects produced by HPF1 on PARylation of histones catalyzed by PARP1 and PARP2 were similar (Fig. 2c, d). At the same time, there was a significant difference between PARP1 and PARP2 in the relative levels of auto- and heteroPARylation determined as % of total ADP-ribose covalently bound to PARP and histones (Fig. 2e, f). While the preferred target of PARP1-catalyzed modification at various concentrations of HPF1 was PARP1 itself (Fig. 2e), most of the PAR synthesized by PARP2 was attached to histones (Fig. 2f). Additionally, the ratio of modification levels of histones and PARP1 decreased with increasing concentration of HPF1, while the respective value in the case of PARP2 was stable when reached ~75%.

To confirm the tendency of PAR chain synthesized by PARP1/ PARP2 to be shortened in the presence of HPF1 as shown previously[31] and here (Fig. 2a, b), we analyzed the length of PAR

synthesized in the absence or presence of three different concentrations of HPF1. Results of autoradiographic analysis of bulk PAR (Supplementary Fig. 1; Supplementary Note 1) show that size distribution of PAR synthesized by both PARP1 and PARP2 was affected by HPF1 in a concentration-dependent manner: increasing the HPF1 concentration enlarged the relative amount of small PAR fraction (1‒10-mers). The concentration dependence of the effect was more pronounced in the case of PARP2, especially when NCP was used to activate the PARylation reaction: the presence of HPF1 at the highest concentration enhanced switch of PARP2 activity to mono(ADP-ribosyl)ation reaction.

**HPF1 promotes PARP1/PARP2-catalyzed PARylation in the absence of DNA and at the excessive DNA concentration**. A transient (low affinity) direct interaction of PARP1 with HPF1 proposed previously[22], prompted us to hypothesize possible stimulation of PARP1- and PARP2-catalyzed autoPARylation by HPF1 in the absence of any activator (NCP or DNA). To verify this hypothesis, the DNA-independent activity (i.e., in the absence of NCP or DNA) of PARP1 and PARP2 was tested in the presence of increasing concentrations of HPF1. Preliminary experiments performed with and without DNAse treatment revealed PARP1, PARP2, and HPF1 preparations being free of DNA contamination (Supplementary Fig. 2; Supplementary Note 2). Low levels of the basal activity detected for both enzymes in the absence of DNA were nevertheless revealed to be increased to different extents in the presence of HPF1 (Fig. 3a, b). The maximal effect on the autoPARylation level of PARP2 was slightly higher (2.4-fold increase) as compared to that of PARP1 (1.6-fold increase). Interestingly, the maximal effect detected for PARP2 was produced by a 4-fold excess of HPF1, while in the presence of DNA or nucleosome, the maximal stimulation was observed at an equimolar ratio of HPF1 and PARP2. The dependence of the optimal HPF1 concentration on the existence of PARP1/2 in the DNA-bound or DNA-free state may result from a higher stability of the ternary complex[22].

Taking into account the capability of HPF1 to stimulate the activities of PARP1 and PARP2 in both the absence and presence of activating DNA, we explored further the PARP1/PARP2-catalyzed automodification reaction in the presence of a high DNA concentration insuring near-complete binding of PARP to DNA, in contrast to conditions used in the experiments described above (see Fig. 2). The maximal stimulatory effects produced by HPF1 on the automodification reaction were observed at a 4-fold excess of HPF1 over PARP1/PARP2 (Fig. 3c). The extent of stimulation detected for PARP2 might be slightly higher as compared to that for PARP1 (3-fold vs. 2.4-fold). Combined, the data obtained in the absence and presence of various DNA concentrations indicate that the HPF1-produced stimulation depends on the relative amounts of PARP in the DNA-free and DNA-bound states.

The previously reported data on modulation of PARP1/PARP2 activity by HPF1[22,24,34] were obtained at high NAD$^+$ concentrations (0.5‒5 mM vs. 1 µM in our experiments). It was therefore important to explore dependence of the HPF1-produced effects on the NAD$^+$ concentration. Preliminary experiments performed for PARP1 (at a fixed PARP1 and HPF1 concentration) in a wide range of NAD$^+$ concentrations revealed that the automodification level was enhanced at low (0.5‒3 µM) and suppressed at high (35‒300 µM) NAD$^+$ concentrations (Supplementary Fig. 4). A detailed study of the effects produced by various HPF1 concentrations on the PARP1/PARP2 automodification level in the presence of 10 µM NAD$^+$ showed maximal stimulating effects at a 2-fold excess of HPF1 over PARP1/PARP2 (Fig. 3d). Notably,

the extent of stimulation detected in these conditions for PARP1 was slightly higher than for PARP2 (2.3-fold vs. 1.4-fold). Thus, our data show that HPF1 can stimulate the PARP1/PARP2-catalyzed automodification over the limited range of NAD$^+$ concentrations.

**HPF1 modulates the auto- and heteromodification activities of PARP1 and PARP2 at the initial stage of the reactions**. Difference between PARP1 and PARP2 in the stimulatory effects produced by HPF1 on their activities in reactions performed over a fixed incubation time, in the varied reaction conditions prompted us to explore kinetics of the auto- and hetero-modification reactions catalyzed by the PARPs. Kinetic measurements of the DNA-dependent activities (in the presence of NCP) in the absence or presence of HPF1 (at the optimal concentration) were performed at 1 µM and 10 µM NAD$^+$ concentrations (Fig. 4a, b). The initial rate of automodification determined to be significantly higher for PARP1 than for PARP2 (13-fold and 70-fold at 1 µM and 10 µM NAD$^+$ concentration, respectively) was enhanced by HPF1 more substantially at low than at higher NAD$^+$ concentration (9-fold vs. 2-fold) (Table 1). The respective value for PARP2 was enhanced significantly at both NAD$^+$ concentrations (11- and 18-fold). As a result, the difference between PARP1 and PARP2 in the initial rates of automodification in the presence of HPF1 was one order of magnitude at the low and higher NAD$^+$ concentration. The initial rate of PARP2-catalyzed histone modification exceeds (1.6-fold) or is comparable with the automodification rate. On the contrary, the PARP1-catalyzed heteromodification reaction is slower than the automodification (~1.8-fold difference in the rates).

Kinetic measurements of the DNA-independent activities of PARP1 and PARP2 were performed in the absence or presence of HPF1 at 1 µM NAD$^+$ concentration (Fig. 4c). Differences between PARP1 and PARP2 in the initial rates of DNA-independent automodification, in the absence and presence of HPF1, were within the experimental error (Table 1). This similarity between PARPs 1 and 2 provides evidence that their basal activities are immediately related to the conserved catalytic domain and can be enhanced to some extent by the direct interaction with HPF1. Comparison of the initial rates of DNA-independent and DNA-dependent modification shows that the DNA-induced activation is more significant for PARP1 (43-fold increase in the absence and 150-fold in the presence of HPF1) than for PARP2 (5- and 11-fold increase in the absence and presence of HPF1, respectively). Notably, the initial rate of automodification in the absence of HPF1 depends on NAD$^+$ concentration more strongly for PARP1- than PARP2-catalyzed reaction (27-fold increase at 10 µM vs. 1 µM NAD$^+$ concentration for PARP1 and 5-fold for PARP2). This difference, which can be explained by a higher activity of PARP1 at the elongation step[10,16], disappeared in the presence of HPF1 (6- and 8-fold increase at 10 µM vs. 1 µM NAD$^+$ concentration for PARP1 and PARP2, respectively). These data suggest that HPF1 controls the balance between initiation and elongation events upon the ADP-ribosylation reaction.

**The extent of HPF1-promoted switch of the substrate specificity depends on the relative PARP and HPF1 concentrations**. To examine how the HPF1-induced stimulation of PARPs activities is related to switching the amino acid specificity of ADP-ribosylation, we explored resistance of the modification to hydroxylamine (HA), which removes specifically ADP-ribose (ADPR) from Asp/Glu residues[25]. The autoradiographic analysis shows that the most portion of ADPR incorporated into PARP1/

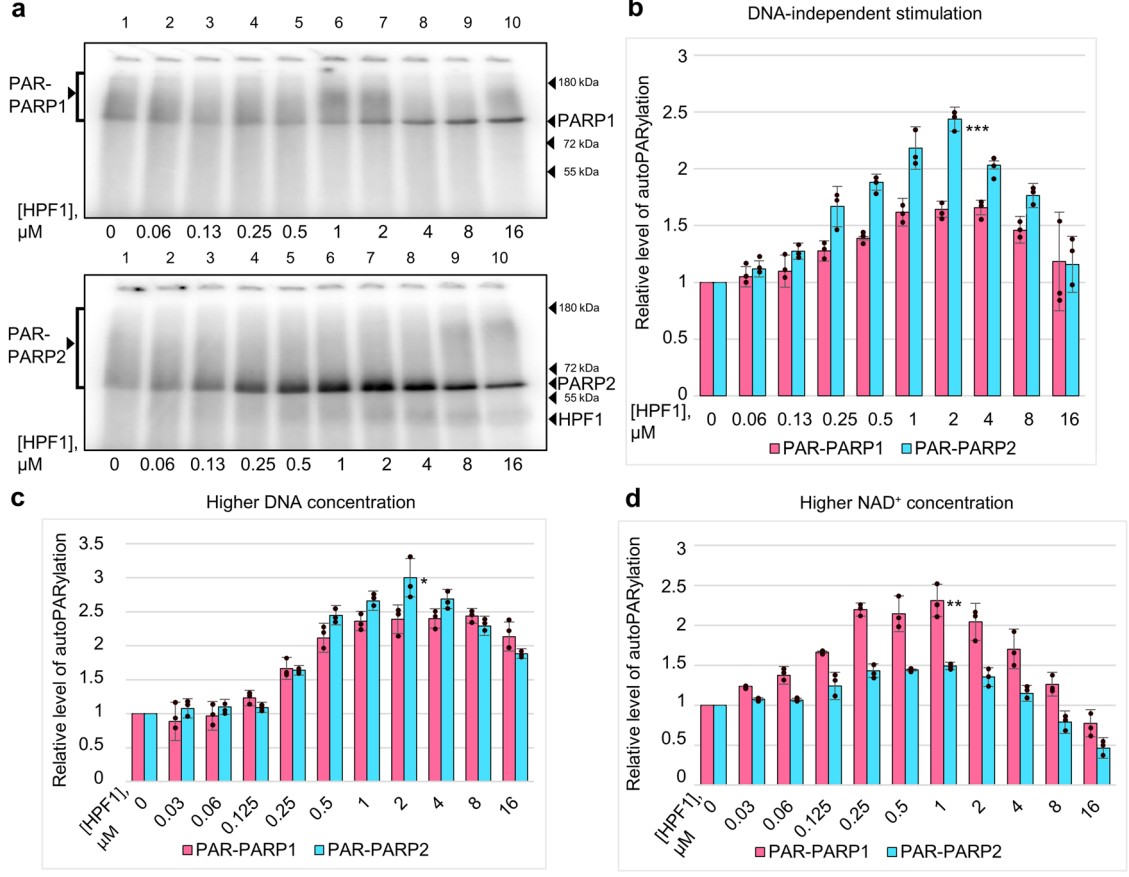

**Fig. 3 HPF1 stimulates PARP1- and PARP2-catalyzed autoPARylation in the absence of DNA and at higher DNA and NAD⁺ concentrations. a** Autoradiograms show covalent binding of $^{32}$P-labelled ADP-ribose to PARP1 and PARP2 after incubation of PARPs (500 nM) with [$^{32}$P]NAD⁺ (1 μM) in the absence (sample 1) and presence of increasing HPF1 concentrations (from 60 nM to 16 μM, samples 2–10) and further separation of products in 20% SDS-PAG. **b** Histograms present relative labelling levels (the mean ± SD of three independent measurements and individual data points) of PARP1 and PARP2 determined by normalization of the ADP-ribose amounts covalently bound to the given protein in the presence of HPF1 to the respective amount in its absence. **c, d** Histograms present relative labelling levels (the mean ± SD of three independent measurements and individual data points) of PARP1 and PARP2 under varied reaction conditions: 2 μM DNA and 1 μM [$^{32}$P]NAD⁺ (**c**); 0.25 μM DNA and 10 μM [$^{32}$P]NAD⁺ (**d**). Original autoradiograms are presented in Supplementary Fig. 3. Statistically significant differences between PARP1 and PARP2 in the effects detected at the optimal HPF1 concentration are marked $p < 0.05$ (*), $p < 0.01$ (**), $p < 0.001$ (***).

PARP2 in the absence of HPF1 was removed by HA, while the loss of modification in the presence of the optimal HPF1 concentration did not exceed 12–20% (Fig. 5a, b). Complete resistance to HA was detected for modification in the presence of a large excess of HPF1. These data indicate that: (1) the extent of switch of the amino acid specificity depends on the relative amount of PARP in the HPF1-bound state; (2) the maximal stimulating effect of HPF1 results from the incomplete switch.

**NAD⁺-hydrolase activity of PARPs 1 and 2 is enhanced by the large HPF1 excess**. The NAD⁺-hydrolase activity, detected for PARP1 in the early study[35], was recently shown to be significantly stimulated in the presence of HPF1[31]. The HPF1-induced switching of PARP1 activity from PAR synthesis to NAD⁺ hydrolysis with formation of free ADP-ribose (up to 90% consumption termed "treadmilling") was observed in the presence of 20-fold excess of HPF1 over PARP1[31]. We were interested to compare the NAD⁺-hydrolase activities at different HPF1 concentrations: optimal for stimulation of ADP-ribosylation reaction and in the large excess (see Fig. 2c, d). The reactions catalyzed by PARP1 and PARP2 activated by NCP were performed without HPF1 and in its presence at the two different concentrations, and the relative amounts of $^{32}$P-labelled ADP-ribose incorporated

into PARPs and histones and released in free form due to NAD⁺ hydrolysis were determined (Fig. 6a-d). In the presence of the optimal HPF1 concentration, the relative amount of free ADP-ribose was lower than in the absence (~5-fold for PARP1 and ~2-fold for PARP2) or at the large excess (~6-fold for PARP1 and ~4-fold for PARP2). Thus, the NAD⁺-hydrolase activity of both PARP1 and PARP2 is enhanced by the large excess of HPF1.

## Discussion

PARP1 is a key nuclear enzyme that catalyzes PAR synthesis as DNA damage response and regulates DNA repair and other processes in cell. It was supposed that PARP1 itself is the most common target of PARylation[36]. At the same time other proteins have been identified as efficient targets of PARylation[37–39]. It is known that PARP1 cooperates with specific partner proteins in numerous cellular processes[40]. Some of these proteins stimulate PARP1 activity and modulate the length of PAR chain[37,41,42]. HPF1 (also known as C4orf27) was discovered recently in the Ahel's group as a cofactor of PARP1, responsible for regulation of ADP-ribosylation signaling in the DNA damage response[21]. Using *C4orf27*$^{-/-}$ cells it was shown that loss of HPF1 induced hyper-autoPARylation of PARP1 and abrogated histone ADP-ribosylation. The capability of HPF1 to limit the level of PARP1

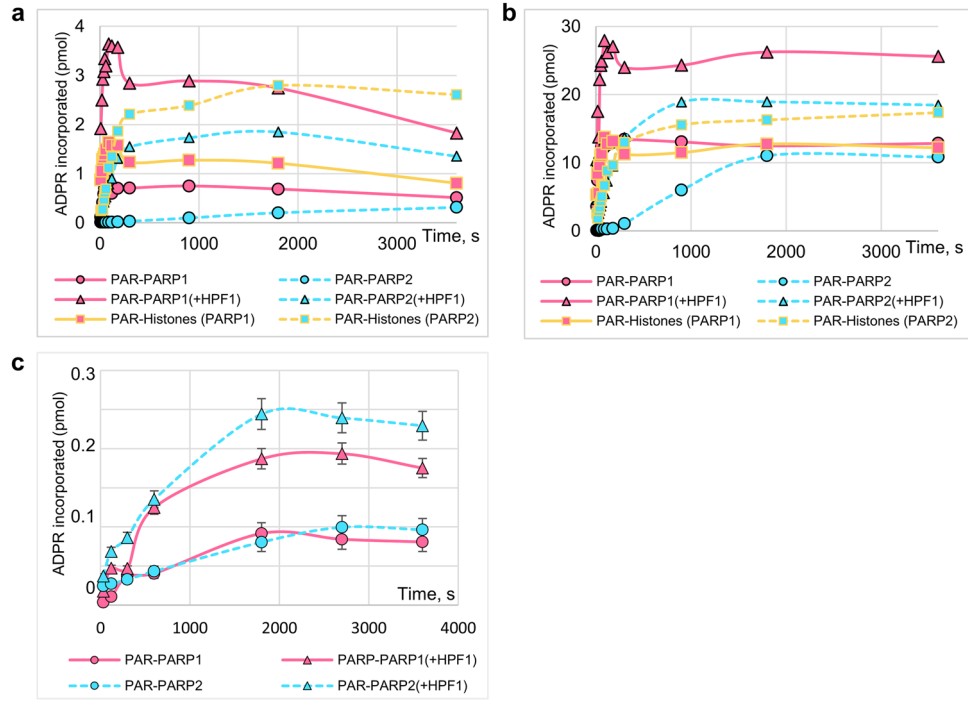

**Fig. 4 HPF1 modulates the auto- and heteromodification activities of PARP1 and PARP2 at the initial stage of the reactions.** Typical kinetic curves of PARP1- and PARP2-catalyzed reactions of ADP-ribose (ADPR) incorporation into PARPs and histones upon incubation of PARPs (500 nM) with [$^{32}$P]NAD$^+$ (**a, c** 1 µM; **b** 10 µM) in the absence and presence of HPF1 (**a, b** 1 µM/0.5 µM in PARP1/PARP2-catalyzed reactions; **c** 1 µM), in the presence of 250 nM NCP (**a, b**) or its absence (**c**). Original autoradiograms are presented in Supplementary Fig. 5.

automodification via shortening of PAR chain can be explained by the interaction of HPF1 with His residue (His826 and His381 in PARP1 and PARP2, respectively)[22], which is critical for PAR chain elongation[43]. On the other hand, the Luger's group showed that HPF1 promotes the NAD$^+$-hydrolase activity of PARP1, which can lead to depletion of NAD$^+$ and thereby prevent production of the long PAR molecules[31]. The authors suggest that the lack of readily available substrate for PARylation due to HPF1-blockage of the PAR chain elongation site results in the increased hydrolase activity.

Previous data demonstrating HPF1-induced modulation of PARP activity were obtained at the large excess of HPF1 relative to PARP1/2[21,31]. At the same time the PARP1 concentration in HeLa cells is significantly higher than the HPF1 concentration (~2030 vs ~104 nM)[21]. Therefore, it was reasonable to explore the influence of varied concentration of HPF1 on the activity of PARP1 and PARP2 in the autoPARylation and histone modification. First, we have found that PARylation in the presence of increasing HPF1 concentration results in shortening of the chain length of PAR covalently bound to PARPs and histones. Interestingly, this effect was observed at both low and high NAD$^+$ concentration (Fig. 2a, b; Supplementary Fig. 1).

Further, we found that the inhibitory action of HPF1 on the PARP1 automodification demonstrated previously[31] depends on the experimental conditions. At a low NAD$^+$ concentration we used in most experiments, the 32-fold excess of HPF1 over PARP1/PARP2 abolished the stimulatory effect detected at the optimal concentration (Fig. 2c, d). On the other hand, at the high NAD$^+$ concentration (35–300 µM) the automodification activity of PARP1 was substantially suppressed even by the 2-fold excess of HPF1 (Supplementary Fig. 4). The affinity of HPF1 for PARP1/2 is very low: the equilibrium dissociation constant (K$_d$) determined for the PARP2-HPF1 complex is in the micromolar range

(>3 µM)[34], and a less stable PARP1-HPF1 complex was undetectable[24]. The interaction of HPF1 with PARPs is stabilized in the ternary complex with nucleosome: the K$_d$ values of 790 nM and 280 nM were determined for the HPF1-PARP1-Nuc165 and HPF1–PARP2–Nuc165 complexes[33,34]. These high K$_d$ values justify high concentrations of HPF1 and its large excess over the PARPs concentration in previous publications[21,31]. Here we revealed that: (1) HPF1 stimulates PARylation of histones even at limited (~8-fold lower) concentration relative to that of PARPs 1 and 2; (2) the maximal stimulating effect on the autoPARylation of PARPs in the presence of both DNA and nucleosome was observed at the equimolar (2-fold excessive) concentration of HPF1 relative to the PARP2 (PARP1) concentration (0.5 µM) (Fig. 2c, d). The lower value of optimal HPF1 concentration determined for PARP2 correlates with the higher stability of the respective ternary complex (as described above). We assume that interaction of HPF1 with PARPs is more efficient in the catalytically active complex due to opening of the HD domain promoted by the NAD$^+$ binding in the active center of PARPs[30]. Probably, accommodation of NAD$^+$ in its binding pocket dynamically alters structure of the PARP-nucleosome complex and facilitates HPF1 binding. The HPF1-induced stimulating effect on the PARPs activities was higher in the presence of nucleosome than model DNA, while the concentrations of HPF1 optimal for the stimulatory action were nearly the same, independently of the type of DNA activator. Possibly, the nucleosome-induced rearrangement of CAT is more suitable for the accommodation of HPF1 and NAD$^+$ substrate[30]. The PARP1 active site interacts with HPF1 in the binary complex of HPF1 with CAT deprived of HD (co-crystallized in the absence of NAD$^+$)[24,43], indicating the necessity of CAT rearrangement for mutual accommodation of HPF1 and NAD$^+$. Taken together, these data suggest the key role of the degree of HD unfolding for

**Table 1 Initial rates of PARP1(PARP2)-catalyzed auto- and heteromodification with ADP-ribose.**

| PARP | DNA-dependent reaction | | | | | | | | DNA-independent automodification | |
| --- | --- | --- | --- | --- | --- | --- | --- | --- | --- | --- |
| | Automodification | | | | Modification of histones | | | | Without HPF1 | With HPF1 |
| | Without HPF1 | | With HPF1 | | | | | | | |
| | $1\,\mu M\ NAD^+$ | $10\,\mu M\ NAD^+$ | $1\,\mu M\ NAD^+$ | $10\,\mu M\ NAD^+$ | $1\,\mu M\ NAD^+$ | $10\,\mu M\ NAD^+$ | | | | |
| PARP1 | 13 ± 5 | 350 ± 20 | 120 ± 15 | 760 ± 30 | 70 ± 20 | 400 ± 50 | | | 0.3 ± 0.1 | 0.8 ± 0.3 |
| PARP2 | 1 ± 0.2 | 5 ± 1 | 11 ± 2 | 90 ± 10 | 18 ± 4 | 100 ± 20 | | | 0.2 ± 0.05 | 1 ± 0.4 |

The initial rates (fmol/s) determined from a linear portion of the kinetic curves (Fig. 4) represent the mean ± SD of three independent measurements.

the formation of the combined HPF1-PARP active site. In addition, our data show that the regularities of HPF1 influence on the PARPs activity are generally manifested on both NCP and DNA.

We showed that the extent of HPF1-dependent switch of PARylation specificity is determined by the relative amount of PARP1/2 in the HPF1-bound state. In the conditions of HPF1-dependent stimulation, 12–20% of ADP-ribosylation is unstable under treatment with HA (Fig. 5b), indicating modification of Asp/Glu residues by HPF1-free PARP molecules. To switch completely the reaction specificity, a large excess of HPF1 over PARP is required.

The next finding of the present study is the more considerable HPF1-induced modulation of PARP2 activity in comparison with PARP1. The maximal level and the initial rate of PARP2 automodification in the presence of NCP/DNA were enhanced more intensively as compared to those for PARP1 (Fig. 2c, d; Fig. 4a, b; Table 1), and the optimal HPF1 concentration for the stimulatory action was 2-fold lower for PARP2. It is noteworthy that under our experimental conditions, the HPF1-PARP2 was more active in the heteroPARylation than in the automodification: about 80% of the total PAR was bound to histones, at both the optimal and highly excessive HPF1 concentrations, while the maximal relative level of PARP1-catalyzed histone PARylation did not exceed 40% (Fig. 2e, f). The specific function of PARP2 to modify predominantly histones was further evidenced from comparison of the initial rates of PARP1/PARP2-catalyzed auto- and hetero-modification reactions (Table 1). PARP2 can therefore play a key role in the subsequent chromatin decompaction necessary for the recruitment of DNA repair factors to the damage sites. This idea is supported by the data of Bilokapic's work on the ability of the PARP2-HPF1 complex to retain two nucleosomes located near the DSB[29], consistent with observations that PARP2 persists longer than PARP1 at DNA damage sites[16,18,44].

Taking into account a transient direct (not DNA-mediated) interaction of PARP2 with HPF1[22], we explored the HPF1-induced modulation of the PARP1/2 activity in the absence of DNA. Absence of statistically significant difference between PARP1 and PARP2 in the initial rates of DNA-independent automodification, in the absence and presence of HPF1 (Fig. 4c; Table 1), provides evidence that their basal activities are immediately related to the conserved catalytic domain and can be enhanced to some extent by the direct interaction with HPF1.

We have shown that only a large excess of HPF1 over PARPs can switch the activities of enzymes from PAR synthesis to $NAD^+$ hydrolysis. This fact is in agreement with the published data also obtained at the high HPF1 concentration[31]. However, our experiments at various HPF1 concentrations revealed no enhancement of the $NAD^+$-hydrolase activity at the low HPF1 concentration, optimal for stimulation of the PARPs activity in the automodification reaction (Fig. 6c, d). The switch of PARP1 to the $NAD^+$-hydrolase activity persists in E284A-mutation of HPF1: the mutant binds to the PARP1-nucleosome complex, but does not promote heteroPARylation of histones[31]. The authors suggest that the inhibitory effect of HPF1 on synthesis of PAR cooperates with the switch of PARP1 activity to $NAD^+$ hydrolysis. This effect results from shielding of the PARP1 elongation center in the complex with HPF1. As a result, PARP1, still associated with activating DNA, quickly runs out of suitable PARylation sites and uses water as a nucleophile instead[31]. We assume that only complete binding of PARP to HPF1, which can be achieved at the large excess of the cofactor, can promote switch to the $NAD^+$-hydrolase activity. However, the cellular HPF1 concentration was shown to be significantly lower than that of PARP1[21].

The concentrations of HPF1 determined to be optimal for stimulating the PARPs automodification are comparable with

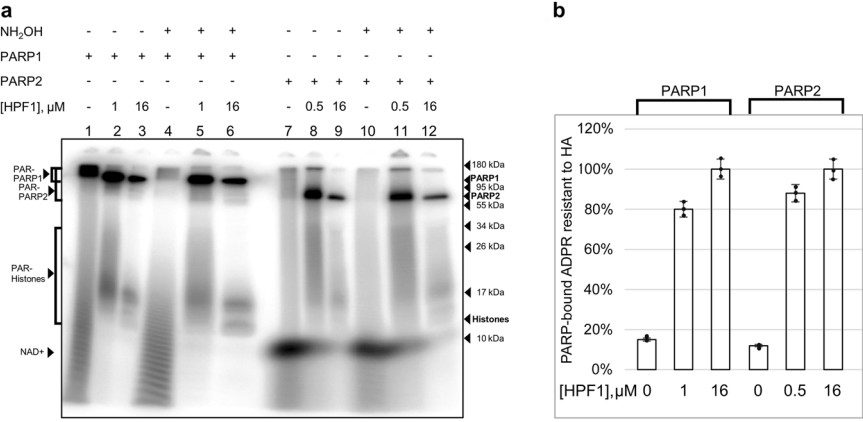

**Fig. 5 The HPF1-induced switch of the amino acid specificity of ADP-ribosylation depends on the HPF1 concentration. a** Autoradiogram shows covalent binding of $^{32}$P-labelled ADP-ribose to PARP1/PARP2 after 30 min incubation of PARPs (500 nM) with [$^{32}$P]NAD$^+$ (1 μM) in the absence and presence of indicated HPF1 concentrations with and without treatment with 1 M HA and further separation of products in 20% SDS-PAG. **b** Histograms present portion of HA-resistant modification (the mean ± SD of three independent measurements and individual data points) determined by normalization of the protein-incorporated ADP-ribose (ADPR) amount in the HA-treated sample to that in the respective untreated sample.

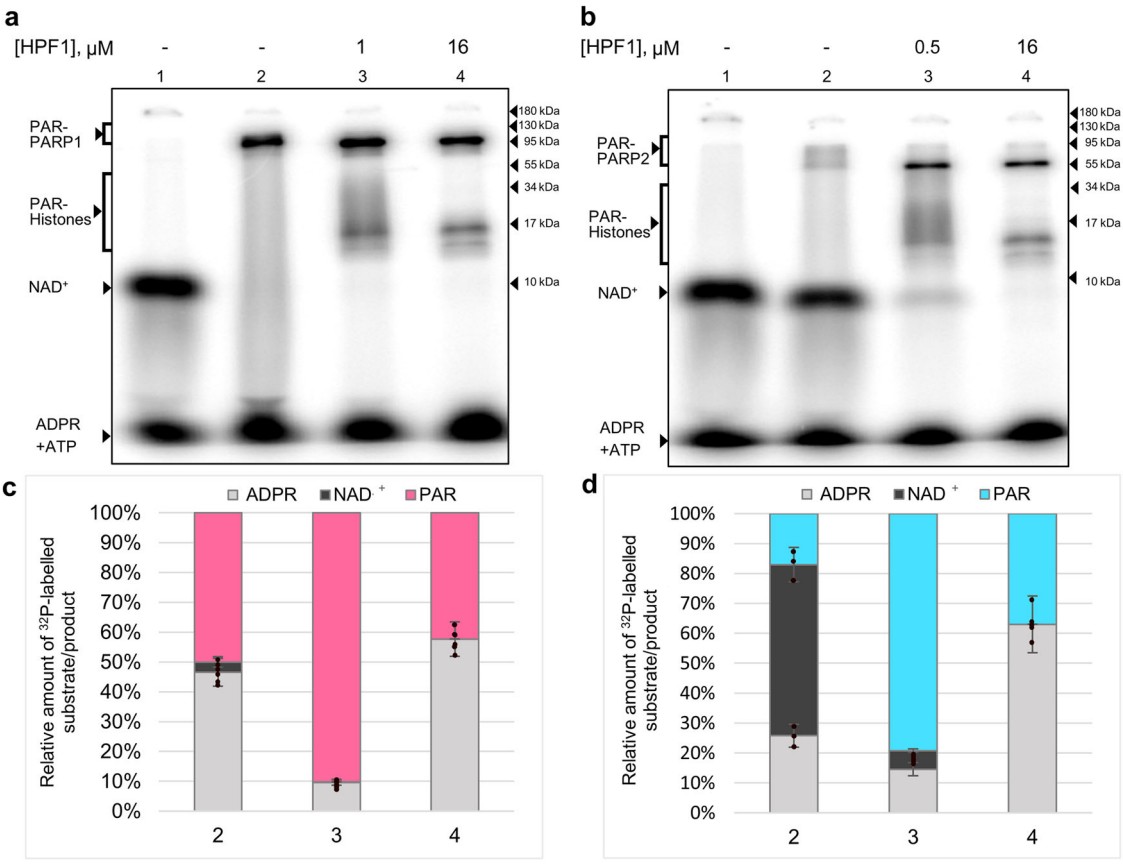

**Fig. 6 The NAD$^+$-hydrolase activity of PARPs is stimulated by the large excess of HPF1. a, b** Autoradiograms of SDS-PAG after separation of ADP-ribosylated PARP1, PARP2, and histones from NAD$^+$ and free ADP-ribose (ADPR). The reaction catalyzed by PARP1/PARP2 (500 nM) in the presence of NCP (250 nM) and [$^{32}$P]NAD$^+$ (1 μM) was performed in the absence or presence of HPF1 (1 and 0.5 μM, PARP1 and PARP2 sample 3, and 16 μM, sample 4). **c, d** Histograms present the relative amounts (%) of [$^{32}$P]NAD$^+$ and $^{32}$P-labelled ADP-ribose covalently bound to PARPs and histones (PAR) and released in free form (ADPR) due to NAD$^+$ hydrolysis in the respective samples. The original [$^{32}$P]NAD$^+$ preparation (control sample 1 without PARP) was contaminated with ATP; its amount was subtracted from the amount of free ADP-ribose (comigrated with ATP) in samples 2–4.

those of PARPs, thus insuring coexistence of the HPF1-PARP complex with free PARP, which can provide sites available for PARylation (Fig. 7a). Taken together, our data allow suggesting the following model of HPF1-dependent stimulation of PARP1/2:

the DNA-bound PARP subunit (1) forms a joint active site with HPF1 (2). The HPF1-free PARP subunit (3) interacts with this complex and serves as a PAR-acceptor. The PARP-HPF1 complex is involved in early stages of serine-specific PARylation,

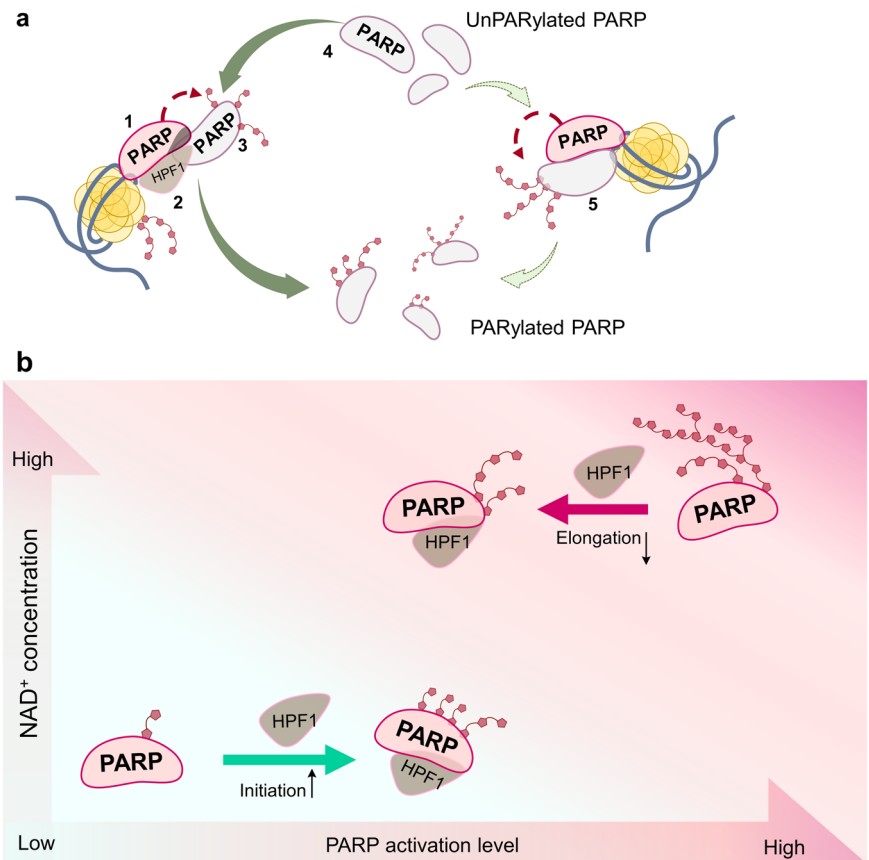

**Fig. 7 Proposed mechanisms of PARP activity regulation by HPF1. a** The HPF1-free PARP molecules serve as available PAR acceptors; their exchange with the PARylated PARP in the complex with HPF1 leads to increase in the PARylation reaction turnover. **b** The HPF1-produced effects on PARP activity depend on the NAD$^+$ concentration: stimulation is predominant function at low NAD$^+$ concentrations, while the hyperactivation at high concentrations of the substrate is suppressed.

which occurs with a high initial rate. The acceptor PARP subunit is PARylated and then substituted by the unPARylated HPF1-free PARP (4); this results in an increased turnover of the whole process. At the same time, the HPF1-free PARP molecules (5) can be non-serine specifically PARylated with a lower initial rate but with a higher elongation level. The high initial rate of HPF1-dependent reaction and presence of the unPARylated PARP excess lead to an increase in the number of acceptor sites, fast turnover of PARPs and removal of the reaction product (due to dissociation of PARylated molecules). Finally, this results in increasing the total level of protein ADP-ribose incorporation.

Data obtained previously[22,24,34] and here under different experimental conditions provide evidence that HPF1 can promote opposite effects on distinct stages of the PARP1- and PARP2-catalyzed reaction: it stimulates early stages and inhibits elongation by shielding of amino acid residues important for PAR chain elongation[22]. Thus, HPF1 can produce different effects on the overall PAR synthesis at low and high NAD$^+$ concentrations (Fig. 7b). Probably, at low NAD$^+$ concentrations the initiation has the greatest contribution to the total reaction output, and the HPF1-induced stimulation of early stages should significantly increase the level of protein-bound ADP-ribose. In contrast, at high NAD$^+$ concentrations, the elongation has a predominant contribution to PAR synthesis. The inhibitory action of HPF1 on the elongation step may therefore suppress the PARylation level at higher NAD$^+$ concentrations. Thus, the hyperactivation of PARP in PAR synthesis at high NAD$^+$ concentrations is inhibited by HPF1. At low NAD$^+$ concentrations, the activity of PARP is stimulated by HPF1, with the optimal concentration and extent of

stimulation being dependent on the relative amounts of DNA-free and DNA-bound PARP.

The dual regulatory function of HPF1 may contribute to maintenance of PARP activity at the level required for its function in the DNA damage response signaling and repair, independently on the NAD$^+$ concentrations. The suppression is most likely predominant function of HPF1 at the normal NAD$^+$ concentration in the nucleus (100 μM[45]) to prevent negative consequences of PARP hyperactivation. The stimulatory action of HPF1 might regulate PARP activity under stressed conditions, when the sustained PARP activity leads to NAD$^+$ depletion.

## Materials and methods

**Materials**. Recombinant wild-type human PARP1 and murine PARP2 were expressed and purified as described in detail previously[46]. Core histones were isolated from *Gallus gallus* erythrocytes and purified as described previously[47]. DNA primers for synthesis of Widom603 147 bp DNA and oligonucleotides for preparation of model 31 bp DNA were synthesized in the Laboratory of Biomedicinal Chemistry (ICBFM SB RAS, Novosibirsk, Russia). pGEM-3z/603 was a gift from J. Widom (Addgene plasmid #26658; http://n2t.net/addgene:26658; RRID: Addgene_26658). Proteinase K was purchased from Applichem (USA), DNase I was purchased from Thermo Scientific (USA). The $^{32}$P-labelled NAD$^+$ was synthesized enzymatically according to method described[48], using [α-$^{32}$P]ATP (with specific activity of 3000 Ci/mmol, synthesized in the Laboratory of Biotechnology, ICBFM, Novosibirsk, Russia). NAD$^+$, NH$_2$OH, reagents for electrophoresis and basic components of buffers were purchased from Sigma–Aldrich (USA).

**Preparation of DNA and nucleosome**. The sequences of the oligonucleotides for model DNA were as follows: template—5′-GGAAGACCCTGACGTTCCCAACTT TATCGCC-3′, downstream primer—5′-pAACGTCAGGGTCTTCC-3′, upstream primer—5′-GGCGATAAAGTTGGG-3′. The model 31 bp DNA containing nick was prepared by annealing the upstream primer and the downstream primer to the

template. The mixture was heated at 90 °C and stepwise cooled to 40 °C, incubated for 15 min and cooled to 4 °C. The effectiveness of hybridization was checked by PAGE in nondenaturing conditions. The amplification of Widom603 147 bp DNA and subsequent reconstitution of nucleosomes using the Widom603 sequence[49] were performed as described previously[47]. Briefly, DNA was obtained by PCR (upstream primer —5′-ACCCCAGGGACTTGAAGTAATAAGG-3′; downstream primer—5′-CCCAG TTCGCGCGCCCACC-3′); nucleosome was reconstituted by dialysis of DNA-histones mixture against a gradient of NaCl from 2 M to 10 mM. The homogeneity of the nucleosome sample was analyzed by the electrophoretic mobility shift assay on a 4% nondenaturing PAG.

**HPF1 cloning, expression, and purification**. To obtain recombinant human HPF1 by expression in *E. coli* cells, a plasmid was constructed. The HPF1-coding sequence was amplified by PCR using specific primers and total HeLa cDNA. The resulting PCR product was annealed with the linearized pLate31 vector (Thermo Scientific, USA). The amplified DNA plasmid was characterized by the Sanger sequencing method at the SB RAS Genomics Core Facility (ICBFM SB RAS, Novosibirsk, Russia). Next, *E. coli* Rosetta (DE3) cells were transformed with the pLate31-HPF1 plasmid. The transformed cells were incubated in a Studier autoinduction system in a 1 l of culture. The growth was carried out for 18 h at 18 °C. Further, cells were harvested and lysed, and the HPF1 protein was purified by sequential chromatographies on the Ni-NTA agarose column (GE Healthcare Life Sciences, USA), MonoQ 5/50 column (GE Healthcare Life Sciences, USA), and Superdex 16/600 column (GE Healthcare Life Sciences, USA). The protein concentration was determined spectrophotometrically, using the adsorption coefficient of the protein based on the Expasy Protparam Data.

**Testing of PARP activity in the poly(ADP-ribose) synthesis**. Catalyzed by PARP1 and PARP2 autopoly(ADP-ribosyl)ation and covalent labelling of histones were carried out in a standard 10 μl reaction mixture containing 50 mM Tris-HCl, pH 8.0, 50 mM NaCl, 5 mM MgCl$_2$, 1 μM [$^{32}$P]NAD$^+$ (or varied NAD$^+$ concentrations specified in Figure legends), 250 nM DNA (nucleosome), 500 nM PARP1 (PARP2), and 0.06–16 μM HPF1. When indicated, the reaction mixture contained no DNA. The reaction was initiated by adding [$^{32}$P]NAD$^+$ to a protein-DNA mixture preassembled on ice. After incubating the mixtures at 37 °C for 15 min for PARP1 and 45 min for PARP2, the reactions were terminated by the addition of SDS-PAGE sample buffer and heating for 3 min at 95 °C. Where indicated, reactions were treated with 1 M hydroxylamine (NH$_2$OH, pH 7.5) for 1 h at 37 °C before addition of loading buffer. The reaction products were separated by 10% (20%) SDS-PAGE (a ratio between acrylamide and bis-acrylamide of 99:1); bands of proteins labelled with [$^{32}$P]ADP-ribose were analyzed by using the Typhoon imaging system (GE Healthcare Life Sciences) and Quantity One Basic software (Bio-Rad). The radiolabelled signals of modified proteins were quantified as follows: the total (raw) signal of the smeared band of modified protein (indicated for each protein in the autoradiograms) was quantified and the same-size background signal of gel in the respective lane was subtracted from the raw signal. The quantitative data presented in histograms were obtained in at least three independent experiments.

**Kinetic measurements of PARP activity**. PARP1/PARP2-catalyzed reaction was carried out in a 70 μl reaction mixture containing 50 mM Tris-HCl, pH 8.0, 50 mM NaCl, 5 mM MgCl$_2$, 1 μM or 10 μM [$^{32}$P]NAD$^+$, 250 nM nucleosome, 500 nM PARP1 (PARP2), and 1 μM (for PARP1) or 0.5 μM (for PARP2) HPF1. Reactions were initiated by addition NAD$^+$ (5 μl) and stopped by dispensing 5 μl aliquot of the mixture to the SDS-PAGE sample buffer, using Multipette E3 dispenser (Eppendorf) for the repetitive dispensing every 10 seconds (until first 60 s of the reaction); the aliquots of longer incubation were dispensed manually.

**Testing of NAD$^+$-hydrolase activity of PARPs**. The reaction of NAD$^+$ hydrolytic consumption catalyzed by PARP1 and PARP2 was carried out in a standard 10 μl reaction mixture containing 50 mM Tris-HCl, pH 8.0, 50 mM NaCl, 5 mM MgCl$_2$, 1 μM [$^{32}$P]NAD$^+$, 250 nM nucleosome, 500 nM PARP1 (PARP2), and two different concentration of HPF1: 1 μM (for PARP1) or 0.5 μM (for PARP2) and 16 μM. After incubating the mixtures at 37 °C for 15 min for PARP1 and 45 min for PARP2, the reactions were terminated by the addition of SDS-PAGE sample buffer and heating for 3 min at 95 °C. The reaction products (covalently bound to PARP1/2 ADP-ribose, unconsumed NAD$^+$ and free ADP-ribose) were separated by 20% SDS-PAGE (a ratio between acrylamide and bis-acrylamide of 99:1) and their yields was analysed by phosphorimaging and quantified as described in the subsection "Testing of PARP activity in the poly(ADP-ribose) synthesis". The quantitative data presented in histograms were obtained in three independent experiments.

**Statistics and reproducibility**. All experiments were repeated three times. Data are presented as mean values ± SD. The *t*-test was used for the statistical analysis. Significant levels are: *$p < 0.05$; **$p < 0.01$; ***$p < 0.001$.

**Reporting summary**. Further information on research design is available in the Nature Research Reporting Summary linked to this article.

## Data availability
The pLate31-HPF1 plasmid constructed for cloning and expression of recombinant human HPF1 protein has been deposited into the database at https://www.addgene.org/; the accession number is Plasmid #176513. Uncropped autoradiograms are provided in the Supplementary Data file. Source data underlying the graphs and charts presented in the figures are available in the Supplementary Data. All other related data will be available upon reasonable request.

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

## Acknowledgements

We would like to thank the entire laboratory of bioorganic chemistry of enzymes for feedback. We acknowledge Ekaterina A. Belousova for supporting initial stages of this work, and Svetlana N. Khodyreva for guidance in obtaining NCP. The reported study was funded by RFBR according to the research projects № 20-34-70028 (protein purification) and № 20-34-90095 (DNA and nucleosome construction), by RSFP № 121031300041-4 (use of shared equipment for experimental work), and by RSF № 21-64-00017 (PARPs activity testing in various conditions).

## Author contributions

T.A.K. performed the experiments and created the figures. N.A.M. and O.I.L. designed the study. M.M.K., K.N.N., and A.A.U. contributed to the study with protein purification and nucleosome assembling. T.A.K., N.A.M., and O.I.L. analyzed the data and wrote the manuscript. All authors reviewed the results and approved the final version of the manuscript.

## Competing interests

The authors declare no competing interests.
