## [Transparent Peer Review File · Communications Biology]

Reviewers' comments:

Reviewer #1 (Remarks to the Author):

ADP-ribosylation is a post-translational modification that plays a vital role in a variety of biological processes. In the DNA damage response, PARP1 or PARP2 bound to damaged DNA forms a complex with a cofactor called HPF1. Accumulating evidence demonstrates that HPF1 promotes the Ser mono (ADP-ribosyl)ation (MARYlation) of histone proteins by PARP1/PARP2 and suppresses the PARP auto-poly(ADP-ribosyl)ation (PARylation). Recently, it has also been reported that PARP1 has activity as an NAD⁺ hydrolase by binding to HPF1. In this study, the authors performed a biochemical analysis of PARP1/PARP2 activity regulation by HPF1. The results showed that the mode of regulation of PARP activity by HPF1 largely depended on the HPF1:PARP1 ratio and NAD⁺ concentration. HPF1 promoted PARylation of PARPs and histones at low concentrations, while at high concentrations, it acted in a repressive manner. The stimulation of PARylation activity of PARPs by low concentrations of HPF1 was also observed in the absence of DNA.

Although there are several recent reports regarding PARP1 and PARP2 activation by HPF1, this study provides novel and interesting insights into the regulation of PARP activity in response to DNA damage. However, there are several concerns that should be addressed prior to publication.

Specific points

1. In Fig. 2, the authors showed that HPF1 stimulated PARylation, not MARYlation, of PARP and histones at low concentrations. It has been demonstrated that the R239A or E284A mutation in HPF1 inhibits the formation of the joint active sites by HPF1 and PARP (Sun et al, 2021, Suskiewicz et al., 2020). Do these mutations affect this PARylation activity of PARP-HPF1? I am also wondering to what extent Ser becomes the ADP-ribose acceptor in HPF1-dependent PARylation.

2. The authors claim that PARP2 is a preferred partner of HPF1 since the autoPARylation level of PARP2 in the presence of NCP/DNA was enhanced by 8-/5-fold at the optimal HPF1 concentration, while the respective effects observed for PARP1 were lower (6-/3-fold enhancement) (Fig. 2c and d). However, this assay could be greatly affected by the basal activity of the recombinant protein used in the experiment. Indeed, PARP1 appears to be highly active even in the absence of HPF1. Careful interpretation is required.

3. In Figs. 2 and 4, PARylation of PARP1/2 is not apparent because the shifted band or smear of PARPs was not clearly detected. The authors should use other methods to confirm auto-PARylation of PARPs, such as immunoblot using specific antibodies.

Minor point

L62: RAR chain should be PAR chain

Reviewer #2 (Remarks to the Author):

The manuscript "Dual function of HPF1 in the modulation of PARP1 and PARP2 activities" by Kurgina T. A. et al. reports an in vitro characterization of the effect of HPF1 on the catalytic output of PARP1 and PARP2 under several experimental conditions.

This manuscript builds on top of pre-existing knowledge of the PARP1/2-HPF1 system, and provides interesting nuance to the established biochemical understanding. While in its present state the biochemical analysis lacks sufficient depth and characterization, several key additions could significantly improve the manuscript, making it an appealing and interesting addition to the field.

Is the increase in auto-PARylation observed by increased concentrations of HPF1 largely driven by the residue switch from aspartate/glutamate to serine – indicating that serine-ADPr is catalytically more

efficient than aspartate/glutamate-ADPr? How does the relative composition of asp/glu-ADPr and ser-ADPr on PARP1/2 change as the HPF1 concentration is increased (as assessed by, for example, hydroxylamine treatment)? It would be important to evaluate and discuss these results in the context of Langelier et al. (biorxiv, 2021), which recently showed that even sub-stoichiometric amounts of HPF1 relative to PARP1 are sufficient to induce a shift towards serine residue and increase the overall catalytic output.

Rudolph et al., eLife 2021 have recently shown that HPF1 induces a dramatic shift of PARP1 activity from automodification to trans-modification – particularly towards a nucleosome substrate. However, according to the data presented in Fig. 2C, even under high concentrations of HPF1 this shift is not observed and PARP1 itself remains the preferential target for ADPr-ribosylation. According to the data shown, levels of histone ADPr peak at 0.25 μ M of HPF1 and decrease afterwards, while PARP1 automodification peaks at \sim 2 μ M HPF1. How do the authors explain this concerning discrepancy?

In Figure 2A, B the signal obtained from auto-modified PARP1 is largely focused in a single band, while the histone signal remains largely smeared across the gel. Does this reflect an inherent difference between chain length (i.e. predominantly short oligo/mono-ADPr on PARP1/2) or does it reflect a different migration pattern in the gel due to the gel chemistry (i.e. higher resolution power for histones allows different chain length to be well separated, while even relatively large poly-ADPr chains on PARP1 migrate closely)? Antibody staining towards mono vs poly ADPr would be very valuable in this context.

Bonfiglio et al., Cell 2020 have recently shown that in cells mono-ADPr is the predominant form of ADPr upon DNA damage and that HPF1/PARP1 preferentially mono-ADP-ribosylates substrates in *in vitro* reactions, and therefore an *in vitro* investigation on the chain length composition of mono- relative to poly-ADPr is relevant.

Langelier et al. (NAR, 2014) have previously shown that PARP2 - as opposed to PARP1 - is preferentially activated by specific DNA templates, although at the time of publication the contribution of HPF1 to PARP1/2 was not known. Characterizing the effect of different DNA templates to PARP1/2 with and without HPF1 would allow us to revisit those key findings in the context of a more complete biochemical understanding of the system, and is relevant to the main point – comparison of HPF1 effects on PARP1 and PARP2.

The authors observe a DNA-independent activation of PARP1 and PARP2. This observation is interesting but a thorough investigation is required to conclusively establish that the observed activation is significant and not an artifact. In particular, how do the authors exclude the possibility of DNA contamination in the PARP1 and/or HPF1 prep.? Additionally, how does the DNA-independent activation of PARP1/2 compare to the DNA-dependent activation in terms of overall catalytic output?

The authors compare free-ADPr produced by PARP1 and PARP2 in the presence of different HPF1 concentrations to conclude that PARP1/2 hydrolysis occurs only at high HPF1 concentrations. Notably, a major difference between the experiments carried out by the present manuscript and Rudolph et al., eLife 2021, is that the latter evaluates the effects of HPF1 on PARP1 hydrolysis in the presence of nucleosomes. The authors should include the addition of nucleosomes as substrate and compare the results obtained without. In addition, there are several other differences between the reactions carried out by the present manuscript and Rudolph et al (NAD concentration, reaction time, DNA length and concentration), how do these factors influence the HPF1-dependent switch to “threadmilling” activity? In Figure 5A and 5B, what is the lower signal detected in lane1 labelled “ADPr+ATP”? Is the NAD⁺ used undergoing non-catalytic hydrolysis to ADPr and nicotinamide? If so, this is a concerning confounder that should be investigated in depth. Has the corresponding signal obtained in lanes 2 to 4 been normalized relative to the background signal in lane1?

Reviewer #3 (Remarks to the Author):

Recent studies highlight the importance of HPF1 for regulating PARP1 and PARP2 activities in the cell. However, as HPF1 has been proposed to exert different effects (affecting amino acid specificity of ADP-ribosylation; limiting PAR chain length; redirecting ADP-ribosylation from automodification to histone modification) and act on both PARP1 and PARP2, it is unclear how these different roles are reconciled and how they play out at physiologically relevant low HPF1 concentrations. Kurgina et al. addresses some of these outstanding questions with an elegant in vitro study that is a valuable addition to the field. We have the following suggestions for shaping the manuscript into a final form: Major points:

- p. 3, line 81-82: "The data allow us to conclude that PARP2 in comparison with PARP1 is the most preferred partner of HPF1". This conclusion is misleading in our opinion, as it suggests that most HPF1 in the cell is associated with PARP2 rather than PARP1 (e.g. due to higher affinity), and the study does not support that. It is possible that while the effect of HPF1 on PARP2 is a bit larger than on PARP1, it is PARP1 that HPF1 most often works with in the cell. It might be better to stick to a statement that the relative effect of HPF1 in terms of stimulation and encouraging histone heteromodification appears greater for PARP2 than for PARP1. The statement about PARP2 being "the preferential partner of HPF1" comes back in the discussion (p. 10, line 316). Similarly in the abstract the authors refer to "the primary role of PARP2" in the HPF1-dependent PARylation – also here it might be better to describe the findings with a more neutral statement that does not imply primary or secondary roles in cells.

- p. 7, lines 195-196: The authors argue that at high NAD⁺ concentrations HPF1 has a negative effect on PARP activity. Firstly, there seems to be a small mistake in figure reference that is to support this statement: I think the authors mean Supplementary Fig. 1a-b rather than Supplementary Fig. 2 (but please double check that). Also, in Supplementary Fig. 1a gel halves with and without HPF1 are not indicated. A more important concern is that the conclusion about negative effect of HPF1 with high NAD⁺ concentrations is drawn based on the signal that corresponds to very long PAR chains on PARP1. Since the reaction is performed over a fixed duration of time that is relatively long when NAD⁺ is plentiful, high NAD⁺ results in very highly modified PARP1. On the contrary, low NAD⁺ concentration results in the same time span in modification of PARP1 with mono ADPr or short/medium chains. In our view it is difficult to compare these different types of modification with each other and conclude that this reflect different behaviour of HPF1 at different NAD⁺ concentrations. It is conceivable and likely in the light of available data that at any NAD⁺ concentration the reaction would be stimulated by HPF1 in the initial phase when the first attachments to the protein are being made and inhibited in the later phase when the chain is being hyperelongated. In such a model, the difference between low and high NAD⁺ concentrations would just consist in how long these different phases take (probably seconds rather than minutes for very high NAD⁺) and whether they are reached within the time point arbitrarily chosen by the authors. To test if HPF1 differently affects different stages of the reaction with high NAD⁺, it would be necessary to conduct a time course experiment in which one could see both the initial and later phases of the reaction of PARP1 in the presence of high NAD⁺ concentrations, with and without HPF1. In the light of these considerations and the result of the time-course experiment, one could also revise the statements in the last paragraph of the discussion where a model with different behaviour of HPF1 at low and high NAD⁺ levels is proposed (p. 10, lines 323-333). Possibly, the opposite effects of HPF1 on early and late stages of the PARylation reaction could be discussed instead. Also, it is worth bearing in mind that in the cell, chain elongation would be counteracted by PARG and ARH3, so the initial phases of the reaction might be relatively more important, even for high NAD⁺, than in vitro – please discuss.

Minor points:

- p. 2, Fig. 1: Panels a and b shows PARP automodification occurring in trans (i.e. one PARP molecule modifying another PARP molecule), while it has been suggested (Eustermann et al., Mol Cell, 2015) that at least for PARP1 it occurs mainly in cis (i.e. the same PARP molecule is both catalysing and receiving the modification). It might be advisable to adjust the figure to reflect this or to comment on both possibilities and respective evidence. In panel c, DNA-dependent heteromodification of 'target proteins' (unlike that of histones in panel d) is shown to be HPF1-independent. Again, this might

provide a biased picture, as recent results point to most target proteins being modified on serine residues and thus most likely in the HPF1-dependent manner.

- p. 2, line 41: The sentence describes the dispensability of PARP1 or PARP2 for cell survival (on the cellular level) but embryonic lethality (on the organism level) of a double knockout. It would be better to refer to the same level in both instances.

- p. 2, line 52: "..., as well as these enzymes cooperate" – change "as well as" to "and" to improve grammar.

- p. 2, line 57: "HPF1 plays an essential role in the PARP1- and PARP2-catalyzed PARylation of histones^{22,24}". Please add another citation here: DOI: 10.7554/eLife.34334.

- p. 3, lines 62-64: The sentence "The authors explained this effect by shielding of important for elongation of RAR chain amino acid residues (His826 and His381 in PARP1 and PARP2, respectively) in the complex with HPF1" could be corrected to: "The authors explained this effect by shielding of amino-acid residues important for PAR chain elongation (His826 and His381 in PARP1 and PARP2, respectively) in the complex with HPF1".

-p. 3, first paragraph: It would be important to introduce here a very recent data showing that PARP1 efficiently extends the mono-ADP-ribosylated sites without HPF1 (DOI: 10.1016/j.molcel.2021.04.028).

- p. 3, line 71: "Lugers" should be corrected to "Luger's"

- p. 3, line 91: Since quantification of radioactive ADPr signal plays an important role in this study, it might be worth describing how the signal was quantified, especially in cases where it is smeared.

- p. 4, lines 123-124: The sentence "While PARP1 was preferentially PARylated itself..." could be changed to "While the preferred target of PARylation was PARP1 itself...".

- p. 7, line 188: The observed difference between PARP2 and PARP1 in this case (3-fold vs. 2.4-fold) is very small and might be partially within the uncertainty of quantification of smeared signal. It would be better to say that the stimulation of PARP2 "might be slightly higher" rather than that it "is higher".

- p. 8, line 239: "C40rf27" should be corrected to "C4orf27"

- p. 9, line 265-268: It would be useful to provide here not just the information about what molar excess of HPF1 over PARPs was needed for optimal stimulation, but also what absolute HPF1 concentration this corresponded to, because it would be the concentration rather than the excess that would be more relevant in the light of the KDs discussed in the previous sentence. The best would be to provide both pieces of information.

- p. 9, line 276: "independently on" should be corrected to "independently of"

- p. 9, lines 277-280: Here the authors discuss His826 and Leu985 as residues that interact with NAD⁺ and refer to Ruf et al., 1998 as the source. However, the structure described in that study (PDB 1A26) shows PARP1 fragment bound to NAD⁺ analogue (carba-NAD⁺) bound in the ADPr acceptor/elongation position, rather than in the NAD⁺ donor position. It seems to us that the authors mixed up these two possible ADPr/NAD⁺ binding sites with each other. The binding of ADPr in the acceptor position is likely mutually exclusive with HPF1 binding. The binding of NAD⁺ in its donor position, which is relevant for the authors' statement, does not involve His826 or Leu985. We think that one could explain the potential positive effect of HPF1 on NAD⁺ binding to the donor site and vice versa by the fact that both events would stabilise movement/unfolding of HD rather than by referring to His826 or Leu985.

- p. 10, line 320 – "Bilokapich's" should be corrected to "Bilokapic's"

Point-by-point response to reviewers

Reviewer 1, comment 1) *In Fig. 2, the authors showed that HPF1 stimulated PARylation, not MARylation, of PARP and histones at low concentrations. It has been demonstrated that the R239A or E284A mutation in HPF1 inhibits the formation of the joint active sites by HPF1 and PARP (Sun et al, 2021, Suskiewicz et al., 2020). Do these mutations affect this PARylation activity of PARP-HPF1? I am also wondering to what extent Ser becomes the ADP-ribose acceptor in HPF1-dependent PARylation.*

The primary aim of our study was to explore HPF1-induced modulation of PARP1 and PARP2 activities in a wide range of HPF1 concentrations. The concentration dependence was not explored previously, but is important for understanding functional roles of wild-type HPF1, which is significantly less abundant in cell compared to PARP1. Possible influence of HPF1 mutations on its functional roles is a special issue. We have to note that previous data obtained by Suskiewicz et al., 2020 are some contradictory. The R239A mutation has no influence on the HPF1 affinity for PARP1 but has impact on the ADP-ribosylation, while the E284A mutation shown to increase the affinity does not modulate the PARP1 activity. Concerning the second suggestion, we performed additional experiments using hydroxylamine treatment to explore extent of the amino acid specificity switch at different HPF1 concentrations. The data obtained and their discussion included into revised version of the manuscript (Fig. 5) and discussed (*lines 321-325*).

Comment 2) *The authors claim that PARP2 is a preferred partner of HPF1 since the autoPARylation level of PARP2 in the presence of NCP/DNA was enhanced by 8-/5-fold at the optimal HPF1 concentration, while the respective effects observed for PARP1 were lower (6-/3-fold enhancement) (Fig. 2c and d). However, this assay could be greatly affected by the basal activity of the recombinant protein used in the experiment. Indeed, PARP1 appears to be highly active even in the absence of HPF1. Careful interpretation is required.*

Our conclusion is based on the lower optimal HPF1 concentration for PARP2 as compared to that for PARP1 and on the specific property of PARP2 to modify more efficiently histones than itself in the presence of HPF1 (in contrast to PARP1 catalyzing with higher rate and total yield the automodification and heteromodification reactions). Our interpretation of the data was reconsidered in the revised version of the manuscript (*Lines 327-335*), taking into account additional kinetic data (Fig.4 and Table).

Comment 3) *In Figs. 2 and 4, PARylation of PARP1/2 is not apparent because the shifted band or smear of PARPs was not clearly detected. The authors should use other methods to confirm auto-PARylation of PARPs, such as immunoblot using specific antibodies.*

The shifted bands of PARylated PARP appear under reaction conditions of hyperPARylation, i.e. at very high NAD⁺ concentrations (in a millimolar range). This is evident from our data presented in Supplementary Fig. 1 (Supplementary Fig. 3 in the revised version). The experiments presented in the main text were performed at low NAD⁺ concentrations (in micromolar range). The extent of smearing depends on the reaction conditions and conditions of products separation: smeared bands are more visible in the samples containing no HPF1 (Fig. 2), and when 10% PAG is used for separation (compare Figs 2a and 2b, demonstrating separation in 10% and 20% PAG). We do not agree that immunoblotting is more useful than radioautography for quantitative experiments. The most of the quantitative data on PARylation published in the literature were obtained by using radioactivity due to better resolution of this method in comparison to immunoblotting.

Reviewer #2, comment 1): *Is the increase in auto-PARylation observed by increased concentrations of HPF1 largely driven by the residue switch from aspartate/glutamate to serine – indicating that serine-ADPr is catalytically more efficient than aspartate/glutamate-ADPr? How does the relative composition of asp/glu-ADPr and ser-ADPr on PARP1/2 change as the HPF1 concentration is increased (as assessed by,*

for example, hydroxylamine treatment)? It would be important to evaluate and discuss these results in the context of Langelier et al. (biorxiv, 2021), which recently showed that even sub-stoichiometric amounts of HPF1 relative to PARP1 are sufficient to induce a shift towards serine residue and increase the overall catalytic output.

First, we would like to emphasize that according to our data the stimulatory effect of HPF1 on the PARP1- and PARP2-catalyzed automodification and heteromodification found to be maximal at an optimal HPF1 concentration decreased when HPF1 concentration was further increased. We have performed experiments using hydroxylamine (HA) treatment. These data (Fig. 5) show that the complete switch of the amino acid specificity to Ser is achieved at the large excess of HPF1, while at the optimal concentration 12-20% of modification remains resistant to HA. We can conclude that the HPF1-induced stimulation results mainly from the catalytically more efficient modification of Ser residues in the joint HPF1-PARP active site and partially by the presence of free PARP capable of catalyzing elongation.

comment 2): Rudolph et al., eLife 2021 have recently shown that HPF1 induces a dramatic shift of PARP1 activity from automodification to trans-modification – particularly towards a nucleosome substrate. However, according to the data presented in Fig. 2C, even under high concentrations of HPF1 this shift is not observed and PARP1 itself remains the preferential target for ADP-ribosylation. According to the data shown, levels of histone ADPr peak at 0.25 μ M of HPF1 and decreases afterwards, while PARP1 automodification peaks at ~ 2 μ M HPF1. How do the authors explain this concerning discrepancy?

From the results presented in Fig. 3a [Rudolph et al., eLife, 2021] no dramatic switch of PARP1 activity to a nucleosome substrate is obvious. The band of PAR-PARP1 seems to be more intensive in comparison with the band of PARylated histones. Moreover, the signal of modified histones appears in the region between the NAD^+ and ADP-ribose signals that significantly increase the background of the autoradiograph. The conditions of separation used in this study are less suitable than our conditions, enabling to separate well-modified histones from NAD^+ and ADP-ribose. In addition, a 5-fold higher PARP1 concentration in our experiments exceeds the concentration of histones and may result in the preferential automodification. The high PARP1 and PARP2 concentrations are used in our study to compare the two PARPs in identical concentration conditions, and the activity of PARP2 is reliably detected and quantified at this concentration. Different optimal HPF1 concentrations for PARP1 and histones can be explained by the increased prevalence of PARP1 as the target at 1 μ M HPF1 concentration.

comment 3): In Figure 2A, B the signal obtained from auto-modified PARP1 is largely focused in a single band, while the histone signal remains largely smeared across the gel. Does this reflect an inherent difference between chain length (i.e. predominantly short oligo/mono-ADPr on PARP1/2) or does it reflect a different migration pattern in the gel due to the gel chemistry (i.e. higher resolution power for histones allows different chain length to be well separated, while even relatively large poly-ADPr chains on PARP1 migrate closely)? Antibody staining towards mono vs poly ADPr would be very valuable in this context.

On the one hand, modified histones with different chain length are more efficiently resolved by PAGE since the protein migration depends linearly on the logarithm of molecular weight, which is significantly higher for PARP1/2. To clarify this point, we have added positions of molecular weight markers to the autoradiograms. On the other hand, in our reaction conditions (high PARP and limited NAD concentrations), predominant modification of PARP1 in the absence of HPF1 is oligo- and mono(ADP-ribosyl)ation due to self-regulation of the ADP-ribose elongation shown in our previous study [Moor et al., Biochimie, 2020].

comment 4): Bonfiglio et al., Cell 2020) have recently shown that in cells mono-ADPr is the predominant form of ADPr upon DNA damage that and that HPF1/PARP1 preferentially mono-ADP-ribosylates substrates in *in vitro* reactions, and therefore an *in vitro* investigation on the chain length composition of mono- relative to poly-ADPr is relevant.

Bonfiglio et al. [Cell 2020] have not shown preferential mono-ADP-ribosylation in vitro. As written on page 1094, “Although we observed an HPF1-dependent shortening of poly-ADPr chains (Figure S7G), as expected (Gibbs-Seymour et al., 2016), we did not detect any PARP1 mono-ADPr in the presence of HPF1 (Figure 6A). This indicates that HPF1, while directing PARP1 away from the formation of long polymers, **does not per se convert PARP1 into a mono-ADP-ribosylating enzyme**”...“the observed **prevalence of mono-ADPr on serine in WT cells** is largely due to the conversion of polymers to monomers by PARG”. In previous studies, shortening of the PAR chains synthesized in the presence of HPF1 was detected by others and explained by shielding of amino acid residues important for PAR chain elongation. These data are described in our Introduction (paragraph 3). To confirm the tendency of PAR chain synthesized by PARP1/PARP2 to be shortened in the presence of HPF1, we analyzed the length of PAR synthesized in the absence or presence of three different concentrations of HPF1. These results are described in the first section of Supplementary material.

comment 5): Langelier et al. (NAR, 2014) have previously shown that PARP2 - as opposed to PARP1 - is preferentially activated by specific DNA templates, although at the time of publication the contribution of HPF1 to PARP1/2 was not known. Characterizing the effect of different DNA templates to PARP1/2 with and without HPF1 would allow us to revisit those key findings in the context of a more complete biochemical understanding of the system, and is relevant to the main point – comparison of HPF1 effects on PARP1 and PARP2.

We agree that this is an important issue. The difference between PARP1 and PARP2 in their requirements to the structure of activating DNA was shown and quantitatively characterized in previous studies of our lab also. These data are briefly described and referenced in the Introduction (paragraph 2) and was summarized in the review (Lavrik O, DNA repair, 2020). As far as “Characterizing the effect of different DNA templates to PARP1/2 with and without HPF1” is concerned, we suppose that investigation of effect of each new DNA template on PARP1/2 with and without HPF1 demands the same detail analysis as it was done in the current manuscript. It is a subject of our further study and is beyond the scope of the present article.

comment 6): The authors observe a DNA-independent activation of PARP1 and PARP2. This observation is interesting but a thorough investigation is required to conclusively establish that the observed activation is significant and not an artifact. In particular, how do the authors exclude the possibility of DNA contamination in the PARP1 and/or HPF1 prep.? Additionally, how does the DNA-independent activation of PARP1/2 compare to the DNA-dependent activation in terms of overall catalytic output?

We have checked absence of DNA contamination in our preparations of PARP1/2 and HPF1. These data have been added to the revised version (the first paragraph of the second section of Results). The data on DNA-independent activity and its stimulation by HPF1 (Fig. 3) were quantified more thoroughly, taking into account the whole smeared band of automodified PARP and subtracting the background signal of gel in each lane. The description of these data has been revised. Additionally, we performed kinetic measurements of DNA-dependent and DNA-independent activities in the absence and presence of HPF1 at the optimal concentration. The kinetic curves are presented in Fig. 4, and the initial rates are summarized in Table of the revised version of the manuscript

comment 7): The authors compare free-ADPr produced by PARP1 and PARP2 in the presence of different HPF1 concentrations to conclude that PARP1/2 hydrolysis occurs only at high HPF1 concentrations. Notably, a major difference between the experiments carried out by the present manuscript and Rudolph et al., eLife 2021, is that the latter evaluates the effects of HPF1 on PARP1 hydrolysis in the presence of nucleosomes. The authors should include the addition of nucleosomes as substrate and compare the results obtained without. In addition, there are several other differences between the reactions carried out by the present manuscript and Rudolph et al (NAD concentration, reaction time, DNA length and concentration), how do these factors influence the HPF1-dependent switch to “threadmilling” activity?

We have performed these experiments in the presence of nucleosomes (Fig. 6). Actually, there is a difference of conditions used in our manuscript and published with Rudolph et al (eLife 2021), such as PARP1 concentration (100 nM vs 500 nM), NCP (200 nM vs 250 nM) and NAD⁺ concentration (200 μM vs 1μM). We suppose that the high concentration of NAD⁺ together with excess of NCP can lead to stimulation of NAD⁺ hydrolysis (Rudolph et al, eLife). In these conditions PARP1 have a limited amount of PAR acceptor sites and excess of NAD⁺. It switches process to the NAD⁺ hydrolysis or so-called “threadmilling” activity of PARP1. We have used low concentration of HPF1 to PARPs keeping in mind their ratio in cell. When we found the conditions for stimulation activity of the both PARPs by HPF1 as well as histone PARylation and showed low level of NAD⁺ hydrolysis at these conditions. We did not extend investigation in-depth of NAD⁺ hydrolysis. It was not a goal of this study. .

comment 8): *In Figure 5A and 5B, what is the lower signal detected in lane1 labelled “ADPr+ATP”? Is the NAD⁺ used undergoing non-catalytic hydrolysis to ADPr and nicotinamide? If so, this is a concerning confounder that should be investigated in depth. Has the corresponding signal obtained in lanes 2 to 4 been normalized relative to the background signal in lane1.*

The original [³²P]NAD⁺ preparation was contaminated with ATP used for the enzymatic synthesis of NAD⁺. To show this contamination and subtract the ATP signal from the total signal of ATP and ADP-ribose (migrated as a single band), we used in each experiment a control sample of NAD⁺ (sample 1 in Fig. 6). To clarify this point, we have revised the figure legend. The last sentence was changed with following one: The original [³²P]-NAD⁺ preparation (control sample 1 without PARP) was contaminated with ATP; its amount was subtracted from the amount of free ADP-ribose (comigrated with ATP) in samples 2–4.

Reviewer 3, comment 1) - p. 3, line 81-82: *“The data allow us to conclude that PARP2 in comparison with PARP1 is the most preferred partner of HPF1”. This conclusion is misleading in our opinion, as it suggests that most HPF1 in the cell is associated with PARP2 rather than PARP1 (e.g. due to higher affinity), and the study does not support that. It is possible that while the effect of HPF1 on PARP2 is a bit larger than on PARP1, it is PARP1 that HPF1 most often works with in the cell. It might be better to stick to a statement that **the relative effect of HPF1 in terms of stimulation and encouraging histone heteromodification appears greater for PARP2 than for PARP1.** The statement about PARP2 being “the preferential partner of HPF1” comes back in the discussion (p. 10, line 316). Similarly, in the abstract the authors refer to “the primary role of PARP2” in the HPF1-dependent PARylation – also here it might be better to describe the findings with a more neutral statement that does not imply primary or secondary roles in cells.*

We agree with the comments. The text of the manuscript has been modified taking into account the comments (line 81-82).

comment 2) - p. 7, lines 195-196: *The authors argue that at high NAD⁺ concentrations HPF1 has a negative effect on PARP activity. Firstly, there seems to be a small mistake in figure reference that is to support this statement: I think the authors mean Supplementary Fig. 1a-b rather than Supplementary Fig. 2 (but please double check that). Also, in Supplementary Fig. 1a gel halves with and without HPF1 are not indicated. A more important concern is that the conclusion about negative effect of HPF1 with high NAD⁺ concentrations is drawn based on the signal that corresponds to very long PAR chains on PARP1. Since the reaction is performed over a fixed duration of time that is relatively long when NAD⁺ is plentiful, high NAD⁺ results in very highly modified PARP1. On the contrary, low NAD⁺ concentration results in the same time span in modification of PARP1 with mono ADPr or short/medium chains. In our view it is difficult to compare these different types of modification with each other and conclude that this reflect different behaviour of HPF1 at different NAD⁺ concentrations. It is conceivable and likely in the light of available data that at any NAD⁺ concentration the reaction would be stimulated by HPF1 in the initial phase when the first attachments to the protein are being made and inhibited in the later phase when the chain is being hyperelongated. In such a model, the difference between low and high NAD⁺ concentrations would just consist in how long these different phases take (probably seconds rather than minutes for very high NAD⁺)*

and whether they are reached within the time point arbitrarily chosen by the authors. To test if HPF1 differently affects different stages of the reaction with high NAD⁺, it would be necessary to conduct a time course experiment in which one could see both the initial and later phases of the reaction of PARP1 in the presence of high NAD⁺ concentrations, with and without HPF1. In the light of these considerations and the result of the time-course experiment, one could also revise the statements in the last paragraph of the discussion where a model with different behaviour of HPF1 at low and high NAD⁺ levels is proposed (p. 10, lines 323-333). Possibly, the opposite effects of HPF1 on early and late stages of the PARylation reaction could be discussed instead. Also, it is worth bearing in mind that in the cell, chain elongation would be counteracted by PARG and ARH3, so the initial phases of the reaction might be relatively more important, even for high NAD⁺, than in vitro – please discuss.

Thank you for the great comments. We fixed figure reference and added indication for gel halves with and without HPF1.

Based on the data obtained previously (Suskiewicz et al, Nature, 2020; Bonfiglio et al, Nat. Comm., 2021), published in Langelier et al. (biorxiv, 2021) and our observations, we agree that HPF1 differently affects the different stages of the reaction. HPF1 stimulates initiation and inhibits elongation (by shielding of amino acid residues important for PAR chain elongation). The opposite effects of HPF1 are revealed at various NAD⁺ concentration. Probably, at low NAD⁺ concentration stimulation of initiation by HPF1 has the greatest contribution in total reaction input and increasing the total amount of synthesized PAR. At high NAD⁺ concentration elongation has a significant contribution in PAR synthesis therefore we observe inhibition of this process by HPF1. It results in decrease of the relative amount of synthesized PAR. Unfortunately, we are not able to distinguish initiation and elongation stages and differential HPF1 influence in our experiments. Moreover, time course experiments under high NAD⁺ concentrations cannot be performed by technical reasons due to high rate of the process.

However, we carried out time course experiments in the presence of 1 and 10 μM NAD⁺ (Fig 4 and Table). This helped greatly expand our research. We show that higher NAD⁺ concentration leads to less amplitude of HPF1-dependent stimulation. These experiments and their discussion included in the revised version of the manuscript (Fig 4, Fig. 7 and lines 378-397).

Minor points:

1) - p. 2, Fig. 1: Panels a and b shows PARP automodification occurring in trans (i.e. one PARP molecule modifying another PARP molecule), while it has been suggested (Eustermann et al., Mol Cell, 2015) that at least for PARP1 it occurs mainly in cis (i.e. the same PARP molecule is both catalysing and receiving the modification). It might be advisable to adjust the figure to reflect this or to comment on both possibilities and respective evidence. In panel c, DNA-dependent heteromodification of ‘target proteins’ (unlike that of histones in panel d) is shown to be HPF1-independent. Again, this might provide a biased picture, as recent results point to most target proteins being modified on serine residues and thus most likely in the HPF1-dependent manner.

The Fig. 1 legend has been modified taking into account the comment; cis and trans mechanisms of autoPARylation are reviewed in the reference added (Alemasova&Lavrik, 2019).

2) - p. 2, line 41: The sentence describes the dispensability of PARP1 or PARP2 for cell survival (on the cellular level) but embryonic lethality (on the organism level) of a double knockout. It would be better to refer to the same level in both instances.

The change suggested has been made (line 44).

3) - p. 2, line 52: “..., as well as these enzymes cooperate” – change “as well as” to “and” to improve grammar.

The change suggested has been made (line 54)..

4)- p. 2, line 57: “HPF1 plays an essential role in the PARP1- and PARP2-catalyzed PARylation of histones22,24”. Please add another citation here: DOI: 10.7554/eLife.34334.

The citation indicated has been added (line 59).

5) - p. 3, lines 62-64: The sentence “The authors explained this effect by shielding of important for elongation of RAR chain amino acid residues (His826 and His381 in PARP1 and PARP2, respectively) in the complex with HPF1” could be corrected to: “The authors explained this effect by shielding of amino-acid residues important for PAR chain elongation (His826 and His381 in PARP1 and PARP2, respectively) in the complex with HPF1”.

The sentence has been corrected (lines 64-65).

6) -p. 3, first paragraph: It would be important to introduce here a very recent data showing that PARP1 efficiently extends the mono-ADP-ribosylated sites without HPF1 (DOI: 10.1016/j.molcel.2021.04.028).

The article specified by the reviewer has been cited by addition of the following sentence: “Recently it was shown that initiation and elongation steps of ADP-ribosylation are distinctly regulated by HPF1, and ARH3 and PARG hydrolases; the HPF1-dependent initial attachment of ADP-ribose to Ser residue can be elongated by PARP1 alone” (lines 68-71).

7) - p. 3, line 71: “Lugers’” should be corrected to “Luger’s”

The correction has been made (line 74).

8) - p. 3, line 91: Since quantification of radioactive ADPr signal plays an important role in this study, it might be worth describing how the signal was quantified, especially in cases where it is smeared.

Description of how protein-bound ADP-ribose signals were quantified has been added to the Materials and methods (lines 445-448).

9) - p. 4, lines 123-124: The sentence “While PARP1 was preferentially PARylated itself...” could be changed to “While the preferred target of PARylation was PARP1 itself...”.

The sentence has been corrected (lines 123-124).

10) - p. 7, line 188: The observed difference between PARP2 and PARP1 in this case (3-fold vs. 2.4-fold) is very small and might be partially within the uncertainty of quantification of smeared signal. It would be better to say that the stimulation of PARP2 “might be slightly higher” rather than that it “is higher”.

11) - p. 8, line 239: “C40rf27” should be corrected to “C4orf27”

The corrections have been made (lines 159 and 276).

12) - p. 9, line 265-268: It would be useful to provide here not just the information about what molar excess of HPF1 over PARPs was needed for optimal stimulation, but also what absolute HPF1 concentration this corresponded to, because it would be the concentration rather than the excess that would be more relevant in the light of the KDs discussed in the previous sentence. The best would be to provide both pieces of information.

The concentration of PARP1/PARP2 has been indicated (line307).

13) - p. 9, line 276: “independently on” should be corrected to “independently of”

The correction has been made (line314).

14) - p. 9, lines 277-280: Here the authors discuss His826 and Leu985 as residues that interact with NAD⁺ and refer to Ruf et al., 1998 as the source. However, the structure described in that study (PDB 1A26) shows PARP1 fragment bound to NAD⁺ analogue (carba-NAD⁺) bound in the ADPr acceptor/elongation position, rather than in the NAD⁺ donor position. It seems to us that the authors mixed up these two possible ADPr/NAD⁺ binding sites with each other. The binding of ADPr in the acceptor position is likely mutually exclusive with HPF1 binding. The binding of NAD⁺ in its donor position, which is relevant for the authors' statement, does not involve His826 or Leu985. We think that one could explain the potential positive effect of HPF1 on NAD⁺ binding to the donor site and vice versa by the fact that both events would stabilise movement/unfolding of HD rather than by referring to His826 or Leu985.

The correction has been made (line316).

15) - p. 10, line 320 – “Bilokapich’s” should be corrected to “Bilokapic’s”

The correction has been made (line337).

REVIEWERS' COMMENTS:

Reviewer #1 (Remarks to the Author):

1. It has been previously reported that HPF1 and PARP1/2 form a joint active site to enable the Ser-ADP ribosylation of histones and limit PARP1 automodification. In Figure 2, the authors claim that HPF1 at physiological concentration could activate both the PARP1 automodification and the heteroPARylation of histones in the complex with the nucleosome. It is an important question whether the novel mode of activation of PARP1 by HPF1 discovered by the authors is similarly mediated by the formation of the joint active site. Since HPF1 R239 is a crucial residue acting to stabilize the local conformation of the HPF1/PARP1 complex and limit PARP1 automodification, examining the effect of the R239A mutation seems to be very informative in this context.

2. In response to the request from Reviewer 3, the authors added a description of a method for the quantification of radioactive ADPr, but it is still unclear how the level of PARylated PARP 1/2 was determined. As noted in the comments and text, the band shift of PARylated PARPs is not evident in reactions with low NAD + concentrations. In these cases, how did the authors determine the area to quantify? For example, do all the bands detected near 180 kDa or in the 55~95 kDa range in Fig. 2a correspond to PARylated PARP1? How was the choice justified?

Reviewer #3 (Remarks to the Author):

The authors have addressed my comments.

Dear Reviewer,

Thank you for consideration of our manuscript. We are grateful to you for useful criticism and suggestions. We have addressed the remaining points raised. Our detailed responses to your comments are as follows.

Reviewer #1 (Remarks to the Author):

Comment 1. It has been previously reported that HPF1 and PARP1/2 form a joint active site to enable the Ser-ADP ribosylation of histones and limit PARP1 automodification. In Figure 2, the authors claim that HPF1 at physiological concentration could activate both the PARP1 automodification and the heteroPARylation of histones in the complex with the nucleosome. It is an important question whether the novel mode of activation of PARP1 by HPF1 discovered by the authors is similarly mediated by the formation of the joint active site. Since HPF1 R239 is a crucial residue acting to stabilize the local conformation of the HPF1/PARP1 complex and limit PARP1 automodification, examining the effect of the R239A mutation seems to be very informative in this context.

First, we would like to emphasize that the HPF1-induced activation was detected at non-excessive HPF1 concentrations insuring a partial binding of PARP1/2 by HPF1 (due to the low binding affinity). We explain the activating effect by coexistence of PARP1/2 in the HPF1-free and HPF1-bound states enabling exchange of the acceptor PARP molecule between these two states as described in detail in Discussion (Fig. 7). This suggestion is further confirmed by an incomplete switch of the ADP-ribosylation substrate specificity in the presence of HPF1 at the optimal for stimulation concentration explored by using hydroxylamine (HA) treatment (Fig. 5). Indeed, a 12–20% loss of PARP1/PARP2 modification (removed by HA) in the presence of the optimal HPF1 concentration, in contrast to ~90% in the absence of HPF1 and 0% in the presence of a large excess of HPF1, indicates that this portion of modification is catalyzed by the HPF1-free PARP1. On the other hand, the most portion of modification at the optimal HPF1 concentration is HA-resistant, indicating that it is catalyzed by the PARP-HPF1 complex. We are therefore very thankful to you for the suggestion to explore the ADP-ribosylation substrate specificity.

Your proposal to examine the effect of the HPF1 R239A mutation interested us. However, it was shown that in contrast to the other HPF1 mutants examined by Sun et al. [Nat. Commun, 12, 2021], the R239A mutant was itself significantly ADP-ribosylated (alongside with a restored automodification of PARP1). Taking into account that the HPF1-induced activation is detectable at a limited NAD⁺ concentration, we do not exclude the inhibiting effect of the HPF1 mutation on the PARP automodification reaction due to the additional NAD⁺ consumption upon the HPF1 mutant modification. Therefore, it is necessary to explore in our reaction conditions several HPF1 mutants shown previously to affect differently the ADP-ribosylation of PARP1/2 [Suskiewicz et al, Nature, 579, 2020; Sun et al, Nat. Commun, 12, 2021]. We plan to expand further our study by using both HPF1 and PARP1 mutants.

Comment 2. In response to the request from Reviewer 3, the authors added a description of a method for the quantification of radioactive ADPr, but it is still unclear how the level of PARylated PARP 1/2 was determined. As noted in the comments and text, the band shift of PARylated PARPs is not evident in reactions with low NAD⁺ concentrations. In these cases, how did the authors determine the area to quantify? For example, do all the bands detected near 180 kDa or in the 55~95 kDa range in Fig. 2a correspond to PARylated PARP1? How was the choice justified?

The low-intensity bands in the 55-95 kDa range represent the proteolytic fragments of PARP1 (co-purified with the full-length protein). Their modification level did not exceed 5% of the total PARP1 modification level, and was not finally taken into consideration. We have revised all presented autoradiograms and indicated precisely for each modified protein the quantified signal region. In the case of PARP1/PARP2, this region starts from the most intense band of mono/oligo-ADP-ribosylated protein (centred on the position of the unmodified full-length protein) and finishes at the beginning of the resolving gel, thus comprising all modified species of various sizes. We apologize for the inaccuracy of the previous design of Figure 2.

Finally, we would like to express our thanks for the helpful comments. We are very much hoping that the revised version of our manuscript will be acceptable for publication.

Sincerely yours,

Olga Lavrik and co-authors